# The Influence of the COVID-19 Pandemic on the Clinical Application of Evidence-Based Practice in Health Science Professionals

**DOI:** 10.3390/ijerph19073821

**Published:** 2022-03-23

**Authors:** Ana Gómez-Sánchez, Carmen Sarabia-Cobo, Cristian Chávez Barroso, Amaia Gómez-Díaz, Concepción Salcedo Sampedro, Elena Martínez Rioja, Ingrid Tatiana Romero Cáceres, Ana Rosa Alconero-Camarero

**Affiliations:** 1Hospital Universitario Marqués de Valdecilla, Servicio Cántabro de Salud, 39011 Santander, Spain; denataychocolate@hotmail.com (A.G.-S.); amagodi82@gmail.com (A.G.-D.); 2IDIVAL Nursing Research Group, Faculty of Nursing, Universidad de Cantabria, 39011 Santander, Spain; alconear@unican.es; 3Hospital Universitario Virgen del Rocío, 41013 Sevilla, Spain; crischavezb@gmail.com; 4Facultad de Enfermería, Universidad de Huelva, 21003 Huelva, Spain; 5Hospital de Día Médico, Hospital Universitario Marqués de Valdecilla en Cantabria, 39011 Santander, Spain; css2256@gmail.com; 6Centro de Atención a la Dependencia Santander, 39011 Santander, Spain; elenamartinez_94@hotmail.com; 7Independent Researcher, 39011 Santander, Spain; tatianacaceres22@hotmail.com

**Keywords:** evidence-based practice, nursing practice, quantitative methodology, survey, non-facultative personnel, barriers, beliefs, attitudes, coronavirus

## Abstract

(1) Background: Evidence-based practice (EBP) informs daily clinical interventions with the purpose of seeking changes to traditional practice through scientific evidence that justifies the reasons for our actions. The objectives were to describe the barriers, beliefs, and attitudes in the application of EBP among university health professionals (not doctors) and to evaluate the influence of the COVID-19 pandemic among them. (2) Methods: This prospective study is both descriptive and observational. The individuals under study were university health professionals (not doctors) from various autonomous regions within Spain, in both public and private spheres. Sociodemographic and labor-related variables linked to the research and its completion were studied. Likewise, the survey instrument Health Sciences Evidence-Based Practice questionnaire (HS-EBP) was administered to evaluate the barriers to, beliefs in, and attitudes towards evidence-based practice. (3) Results: A total of 716 responses were gathered, of which 387 were collected during the period of confinement, and 343 in the COVID-19 post-confinement period. Possible associations that might help respond to the objectives were explored through a correlational study between the sociodemographic variables and each sub-scale of the HS-EBP 30 questionnaire (*n* = 716). (4) Conclusions: Barriers to, beliefs in, and attitudes towards evidence-based practice are described. There is a leadership gap where line management provides insufficient motivation to follow work routines. The COVID-19 pandemic has caused immense stress among health professionals. The post-confinement group showed a significant change in the variables “beliefs and attitudes”, and likewise in the “evaluation” block, justified by the need to update knowledge and to apply evidence.

## 1. Introduction

Currently, we can speak of evidence-based practice (EBP) generically for all health-related disciplines that have adapted their professional practice to the knowledge generated by quality scientific research, incorporating professional experience, the demands and values of patients/users, and existing resources. As a consequence, the decisions that professionals must make to choose the best care for their particular patients, when supported by these elements, lead to less variability in clinical practice, and working with evidence-based practice becomes part of the culture of quality in the health care we provide to our patients. Evidence-based practice guides daily clinical interventions with the end purpose of seeking changes to traditional practice through scientific evidence that justifies the reason for a health care approach. In turn, it seeks the resolution of problems so as to offer health care that integrates the best evidence from published studies that defines data on health and adapts the preferences of the patients. This concept has its origin in evidence-based medicine (MBE), described by Sacket et al. [1] as “the conscientious and judicious use of current best evidence from clinical research in the management of individual patients”.

Experts in EBP, in the declaration of Sicilia, have indicated that all health professionals should understand the basic concepts of EBP and should establish evidence-based protocols. They should, in addition, maintain a critical opinion, not only towards conducting their practice but also towards the results of the investigations. Without these skills, professionals who are implementing these practices will face significant difficulties [2].

EBP is defined as the interest of professionals in knowing the degree of certainty or uncertainty on which they base their care or uncertainty on which the care they provide to their population is based, and to what extent new quality research can increase the evidence of clinical practice [3]. In this way, health professionals integrate the best evidence and achieve more favorable health-related results both from the clinical and the management perspective [3].

According to Worum et al. [4], empowerment within the work environment leads to EBP. Professionals showed greater commitment, creativity, and productivity when they had access to information and the support of line management, sufficient human resources, and sufficient opportunities for work to progress [4].

This work-related philosophy involves all professionals dedicated to a person’s health. However, despite the benefits of EBP, its adoption in practice has been inconsistent in the different areas of professional health care (nursing, physiotherapy, occupational therapy, speech therapy, etc.) [5]. EBP research applied to different collectives is heterogeneous, even within the same professional category. For example, within the field of nursing, EBP is studied more in specialized nursing than in community nursing [6].

According to the World Health Organization (WHO) [7], integrated care achieves better quality care with optimum benefits. Interprofessional collaboration in training, practice, and research processes establishes a priority and a means of both benefiting from group work and reducing disequilibrium and the needs of health care workers. The WHO recognizes that interprofessional training can strengthen collaboration and it is key to the optimization of team members’ skills, the manipulation of cases, and the delivery of better quality health services, achieving better health care results.

Professionals find different barriers in the implementation of EBP [6]. A lack of line management authority to introduce changes in practice, insufficient support for health organizations, insufficient use of the English language, lack of time because of work overload, a scarcity of personnel, personal or family situations, and the knowledge gap or negative beliefs in EBP are some of the barriers highlighted in previous investigations [4,6,8,9]. The process of change is another barrier to the application of the best evidence in clinical practice [5,9].

There is extensive evidence that the findings of the investigation were applied in an inconsistent manner. Firstly, in the United States, 46.4% of nurses thought that EBP was used in a routine way in their clinical practice [10]. In Europe, the data were less inspirational, because only 24% of nurses thought that they applied research results in their daily clinical practice [11]. Furthermore, with respect to the preferences of the patients, there are hardly any current studies with quantitative designs. The scant incorporation of the values and the preferences of patients within practical strategies has been recognized. Both reviews and studies with qualitative designs have concluded on a lack of tests, given that it depends on many factors, and is as yet an unresolved problem [12].

The limitations for this approach to spread in the health care system include the following: (1) Since it is secondary research, the lack of original research of sufficient methodological quality to support it makes it necessary to begin by conducting original research, an essential raw material for evidence-based practice. (2) There are barriers for professionals and institutions that prevent the application of the results in patient care and that must be overcome so that quality care can reach its intended recipients. (3) Resistance to change in health care professionals is another obstacle to overcome. This can be exacerbated by a lack of habit in reading research articles. (4) Some professionals may have certain difficulties in assuming the responsibilities of an autonomous professional, given their previous professional career. (5) In the knowledge society in which we find ourselves, knowledge changes so rapidly that it is necessary for the professional to have skills that allow him or her to keep up to date with research results. The lack of these, as well as the overload of health care work to which they are subjected, often prevents them from reviewing the knowledge acquired during their university training.

These limitations can be overcome by assuming, in the first place and from a critical position, our own barriers and considering, in the second place, that a fundamental part of our professional task is to do research based on our needs and resources. Only in this way is it possible to generate disciplinary knowledge that is also sensitive to the cultural environment in which we find ourselves. It is for all these reasons that our study is justified.

The justification to carry out this Spanish project depends on the academic limitations in existence up until the entry into force, in 2008, of the degree studies adapted to the framework of the European Higher Education Area. Up until that date, the professionals at the core of the present study had followed a university training of three years focused more on health care than on investigative areas. At present, the 4-year degree courses include the development of student competences in the scientific field. This change generates greater training in the field of investigation, but it also gives access to master’s and doctoral programs, previously limited for the professionals who followed alternative study routes, needing a higher dose of motivation and commitment to overcome all possible obstacles. These advances have permitted an increase in scientific activity in the area of nursing; at present, it occupies seventh place in the global ranking of scientific production [13].

According to the existing evidence, the level of academic preparation is directly related to greater knowledge of EBP and greater intervention in practices dependent on scientific support. In the literature review, positive relationships were noted among attitudes, knowledge and skills, and the frequency of the use of EBP among nursing educators within various countries [14]. These facts optimized the implementation of the research [15,16], with a lower perception of barriers and a higher acceptance of EBP [17]. Hence, health professionals within the university have lengthier training (master’s, doctorate), and, therefore, they are the ones with greater knowledge of EBP; paradoxically, health care professionals, whose practice is directly centered on the patients, implement fewer EBP-related procedures [6,18,19].

The development of various tools allows us to identify aspects related to the use of the research results and the application of EBP—attitudes, knowledge, skills, beliefs and values, and practice as well as facilitators and/or barriers that influence its application. These validated instruments of measurement are very advantageous because they can be used to compare very diverse practices within clinical settings through pertinent transcultural adaptations with respect to the original. They also serve to introduce changes in practice based on the findings that have been shown, despite the limitations they present, as other authors have explained [20].

In the present study, the Health Sciences Evidence-Based Practice (HS-EBP) questionnaire was chosen [21], which, unlike others, is the only one that measures all the steps in the EBP process, including the principal external factors that influence it (individuals and organizations). This HS-EBP measurement instrument is of a transdisciplinary nature. It, therefore, permits objective comparisons between the different groups of professionals under study, favoring the implementation of EBP in different clinical settings. It also shortens the time lag between scientific advances and practice in clinical decision-making [22].

The principal objective of this study is to describe the barriers to, beliefs in, and attitudes towards the application of evidence-based practice (EBP) among university professionals of health sciences (excluding doctors and odontologists). In the course of data collection for this work, a state of emergency was declared, and a three-month domestic national lockdown was imposed in Spain, between March and June 2020, due to the COVID-19 pandemic. These events even prompted the enlargement of the study, incorporating a second objective—to evaluate the influence of the COVID-19 pandemic on the barriers to, beliefs in, and attitudes towards EBP among university professionals of health sciences, excluding doctors.

## 2. Materials and Methods

### 2.1. The Study Design

An observational, analytical, and prospective study was proposed.

### 2.2. Individuals under Study

The individuals under study were university health professionals from Spain. Their criteria for inclusion were health professionals from the public and/or private arena; in employment, including recent graduates. The exclusion criteria were doctors, odontologists; over five years with no professional activity; teachers and/or researchers in full-time employment. An intentional convenience sampling of the university of study formed the sample at two very different moments, setting up two groups—responses collected before the lockdown due to COVID-19 (January to June 2020) and post-lockdown (July–September 2020). Participation was voluntary and they were not remunerated in any way.

### 2.3. Study Variables

-Sociodemographic and workplace variables: region, age, sex, profession, academic training, place of work (hospital, primary care, others, public, private, state-supported) and service or unit, employment situation (permanent, internship, temporary), family load, length of service in current employment, and work shifts.-Evaluation of barriers to, beliefs in, and attitudes towards evidence-based investigation. The transdisciplinary Health Sciences Evidence-Based Practice Questionnaire (HS-EBP) was used [21]. This questionnaire has 60 items, with a Likert-type scale classified on the following sub-scales: beliefs and attitudes (D1), consisting of 12 items, with a range of possible scores of between 1 and 120; results of the scientific investigation (D2), with 14 items and a range of possible scores between 14 and 140; development of professional practice (D3), with 10 items and a range of possible scores from 10–100; the evaluation of results (D4) and barriers/facilitators (D5), with 12 items and a range of possible scores between 12 and 20. It is a positive scale, with higher scores on each subscale indicating greater weight on the subscale that was evaluated.

### 2.4. Procedure

A purposive sample selection was made, accepting all the questionnaires that were received. No questionnaires were eliminated since they were all complete. A mass e-mail with a link to the self-administered survey was sent out to professional colleges, hospitals, health centers, private clinics, and social networks. All the variables and the HS-EBP questionnaire (21) were integrated into the Google Forms survey creation online tool. A letter of presentation was prepared with the request for collaboration, explaining the purpose of the study, a description of the questionnaire (including the number of items and the estimated time for their completion), and information on the research team of the project. The questionnaire was designed so that there was only one response option for each mailing. The same was done for the collection of data on the post-COVID-19 lockdown period.

### 2.5. Data Analysis

SAS v9.4 software (SAS Institute Inc., Cary, NC, USA) was used for the statistical analysis. The settings for the statistical analysis took a significance level of 0.05. Bivariant tests were performed between the response variables (5 dimensions and total) and the other variables. The Shapiro–Wilk normality test, Levene’s Homogeneity of Variances test, and other relevant statistical tests (Student-t test/ANOVA and the Mann–Whitney–Wilcoxon/Kruskal–Wallis test) were all applied. A descriptive analysis of all the variables and a Pearson correlation analysis of the sociodemographic variables and the HS-EBP 30 questionnaire were performed. The results of a single-factor ANOVA were used to evaluate possible differences by professional category with respect to the subscales of the questionnaire. Subsequently, an analysis was performed to evaluate the differences between the pre- and post-lockdown groups on the subscales of the HS-EBP 30 questionnaire (single-factor ANOVA). The results are considered to have a significance level of *p* < 0.05.

### 2.6. Ethical Considerations

A letter of presentation with the request for collaboration was drafted. It set out an explanation of the objective of the study, detailing the characteristics of the questionnaire for data collection (number of items and estimated time for completion), information on the research team, and finally, a link to access the questionnaire. A formal request to carry out the study was sent to the management bodies of the various health centers that were participating. Likewise, the Ethics Commission of the Cantabrian Health Service was contacted to request permission for the study (CE 2019.288). The anonymity of all participants in the study was guaranteed following the provisions of Organic Law 3/2018 on the Protection of Personal Data.

## 3. Results

A total of 716 responses were gathered, of which 387 were from the lockdown period and 343 from the post-lockdown period due to COVID-19. In all, 39% of those contacted responded to the second mailing of the questionnaire. In Table 1, the sociodemographic variables are shown for the whole sample.

All variables followed a normal distribution (*p* > 0.05) and variance tests indicated the homogeneity of variances (*p* > 0.05). Possible associations were explored through a correlational study between the sociodemographic variables under study and each subscale of the S-EBP 30 questionnaire (*n* = 716). The results are shown in Table 2. We can see a significant association between the Results, Evaluation, and Barriers subscales with age (the younger the age, the more difficulties and barriers to the implementation of EBP). Significant correlations were also found between the Results subscale with work in primary care (indicating less consumption and implementation of EBP), with having family responsibilities, with having been working for less time (inverse relationship), and with having taken specific training courses (inverse relationship). The Development subscale presented significant associations with time worked (inverse) and with having taken specific courses (also inverse). The Evaluation subscale presented significant correlations with working in primary care (positive relationship), and with years worked (inverse) and having taken specific courses (also inverse). Finally, in the Barriers subscale, significant associations were found with having taken specific training (inverse). We can summarize that people who work in primary care, are younger, with fewer years of experience, and with little or no specific training in EBP, apply scientific evidence less in their clinical practice and have more difficulties in its interpretation and application.

A descriptive analysis on the subscales of the questionnaire was performed with a professional category (Table 3). We can see that the mean values in all the subscales are similar in all professional categories. In the following analysis, we are able to extract the inferential analysis.

The ANOVA analysis identified the professional category as the independent variable and the questionnaire subscales as dependent variables. Statistically significant differences were found in all subscales (see Table 4).

The results of the HS-EBP 30 questionnaire for both groups of the study were compared for each subscale to respond to one of the objectives. In Table 5, the average responses given by the whole sample and for both groups (pre- and post-lockdown due to COVID-19) are shown, as well as the comparative statistical analysis between both groups through the single-factor ANOVA test.

## 4. Discussion

In the present study, a response to the principal objective was given—describing the barriers, the beliefs, and the attitudes that currently exist in the application of EBP. With regard to the secondary objective, after the evaluation of the post-lockdown group, the results show a significant change in the variables related to the two blocks of “Beliefs and Attitudes” and “Evaluation”.

From the 716 survey responses to the question “Have you followed specific training in investigation in the past 5 years?”, significant associations were noted with all subscales of the HS-EBP 30 questionnaire. A majority of speech therapists were prominent in all the subscales of the questionnaire, followed by podologists and then nurses, a fact that was corroborated in the inferential analysis. The typical respondent profile was a nurse, in permanent employment, in a public-sector hospital, with no family responsibilities.

From the results of our study, it may be pointed out that health professionals (excluding doctors) defined EBP as an integral part of their work and expressed a positive opinion towards it, attributing a strong impact to it in the quality of attention offered to the user. As in other studies, the health professionals were motivated to include it in their daily practice [22]. Nevertheless, EBP has not integrated itself into health care due to various factors, although interest and motivation towards research has a growing tendency. The perceived barriers in this study coincided with those found in others, regardless of the country in which the studies were developed—lack of knowledge and sufficient skills to filter and to implement the best scientific results in their area of work, lack of time, lack of support and stimulation of EBP among line managers, as well as a lack of independence to introduce changes within the working environment [6,22].

Personal motivation to improve the nurse–patient relationship and the grade of professional development stood out as facilitating factors for the implementation of EBP [5].

Another facilitator was continuous specific education in research matters. However, the interviewees pointed to both time pressures and workloads as barriers to both following the training and implementing new knowledge. This problem was generalized in all the studies that were reviewed [4,5].

The degree of independence of the professional in the job also affected EBP. In our study, there was a significant difference between the primary health care professionals and those from other areas and levels of health care assistance, among whom the former employed more scientific evidence. Nevertheless, the relationship between the intake of scientific literature and a higher degree of autonomy in the workplace differed from the implementation of EBP. This relationship was more heterogenous in the sample and appeared to respond to individuals more than the collective interest [5]. The fact that health care attention appeared not to be sufficient motivation in itself meant that other strategies were required to generate research questions of group interest and, therefore, to achieve greater commitment towards the EBP implementation process [5].

The empowerment of a hierarchical structure at work through a line manager, teamwork, recognition, and the positive effects of action all facilitated commitment towards EBP [4,5]. Our results show an absence of leadership in relation to the integration of PBE, where the hierarchical line managers neither sufficiently motivate nor sufficiently incentivize the adoption of PBE in work routines. Giménez et al. [23] pointed to the need to support the line managers in research that guides the development of studies among motivated professionals.

The younger nurses and those with greater training reported greater familiarity with EBP, information that is echoed in other studies [24].

The global COVID-19 pandemic has become a source of great stress among health professionals. It motivated us to perform a comparative pre- and post-lockdown study with the aim of showing possible differences with regard to interest in EBP. In our study, the post-lockdown group showed a significant change in the variables related to the two blocks of “Beliefs and Attitudes” and “Evaluation”. There was a significant increase in the interest of interviewees towards these areas of EBP, possibly justified by the need for constant knowledge updates and for applying evidence, given the high uncertainty generated by the virus.

It is essential that the training in EBP is continuous within both the university and the area of clinical health care. Special emphasis must be placed on the removal of barriers, both for access to scientific evidence and its introduction within health care units [6].

Real policies, aware of the importance of EBP, are key to creating an infrastructure that can accept its implementation. So too are policies that provide financing to supply the necessary human and technical resources to achieve EBP, so that a time in the working day may be set aside to move closer to the latest scientific evidence and therefore create the opportunities to learn and to grow at work. If implemented, it will have a waterfall effect on the organization and the internal motivation of its professionals [4], giving access to information and to specific training, as well as to improvements to their understanding of EBP and its implementation in clinical practice. The responsible coordination of EBP is, for this reason, necessary [23].

Integrating EBP into work routines, placing it within reach of all professionals, with leaders and health care policies that support it, could ensure that they are updated and could ensure the possibility of offering scientifically validated care. It is necessary to find adequate tools to reduce the barriers and difficulties that these professionals have mentioned, implementing tutorized working groups, manuals, and specific training plans.

As future research lines, it could be of interest to prepare comparative studies in which professional doctors are included to evaluate the differences that exist between those professional categories. Likewise, the design of appropriate tools can reduce the barriers and difficulties that have been discussed.

## 5. Limitations

A limitation was the heterogeneity of the responses in relation to the participation of the autonomous regions of Spain.

Some sections of the questionnaire received few responses, conditioning some data limited to only a few items.

## 6. Conclusions

In the present study, a series of barriers, beliefs, and attitudes towards the application of evidence-based practice have been described, such as a lack of knowledge, time, support, motivation, lack of independence, and increased workloads that in no way facilitate the completion of continuous training, with a deficit of leadership, where the responsible line managers provide neither sufficient motivation nor sufficient incentivization to follow work routines.

On the contrary, the facilitating factors had personal motivation for the improvement of the nurse–patient relationship, the professional levels that had been reached, and specific continuing education in research areas.

The COVID-19 pandemic has caused high-stress levels among health care professionals; the post-lockdown group showed a significant change in the variable of “Beliefs and Attitudes”, justified by the need to update knowledge and to apply evidence.

It is essential that training in EBP be continuous from university up to clinical health care assistance. Greater emphasis must be placed on the removal of barriers, both for access to scientific evidence and for its introduction within health care units, in which it appears that line management is deficient if not indeed absent. These findings can guide health service managers in planning strategies to improve the level of EBP competence of professionals, broadening efforts (which have traditionally been directed at improving attitude, skills, and knowledge in EBP) to achieve real utilization of EBP. The COVID-19 pandemic has highlighted the importance of knowing how to consume scientific literature and the importance of applying clinical practices based on scientific evidence.

## Figures and Tables

**Table 1 ijerph-19-03821-t001:** Sociodemographic descriptive variables of the whole sample.

Variable	Variable	*n*	Percentage
Gender	Men	94	13.1%
	Women	622	86.9%
Autonomous community	Andalusia	14	2.0%
	Aragon	33	4.6%
	Asturias	7	1.0%
	Cantabria	287	40.1%
	Castilla la Mancha	19	2.7%
	Castilla y León	77	10.8%
	Catalonia	42	5.9%
	Extremadura	21	2.9%
	Galicia	68	9.5%
	Balearic Islands	8	1.1%
	La Rioja	7	1.0%
	Madrid	21	2.9%
	Murcia	21	2.9%
	Navarra	4	0.6%
	Basque Country	74	10.3%
	Valencia	9	1.3%
	Andalusia	14	2.0%
	Aragon	33	4.6%
Family responsibilities	Yes	328	45.8%
	No	388	54.2%
Profession	Nurse	341	46.7
	Physiotherapist	146	20.0
	Speech Therapist	100	13.7
	Podologist	18	2.5
	Occupational Therapist	103	14.1
	Others	22	3.0
Labor situation	Temporary	160	22.3%
	Internship	172	24.0%
	Permanent	384	53.6%
Shift work	Yes	243	33.9%
	No	473	66.1%
Work in hospital		352	49.2%
	Public	313	43.7%
	State-assisted	20	2.8%
	Private	31	4.3%
Works in AP		603	84.2%
	Public	89	12.4%
	State-assisted	5	0.7%
	Private	19	2.7%
Work at another site other than a hospital or AP		405	56.6%
	Public	68	9.5%
	State-assisted	25	3.5%
	Private	218	30.4%

**Table 2 ijerph-19-03821-t002:** Correlational study between sociodemographic variables and subscales of the HS-EBP 30 questionnaire.

Variables	Beliefs and Attitudes	Results	Development	Evaluation	Barriers/ Facilitators
Age	−0.061	−0.115 *	−0.081	−0.106 *	−0.032 *
Gender	0.036	−0.079 *	0.028	0.032	−0.042
Habitual place of work (Hospital)	−0.023	0.054	0.067	0.066	−0.047
Habitual place of work (Primary Health Care)	0.030	0.225 *	0.130	0.294 **	0.125
Habitual place of work (Other)	−0.066	0.031	0.001	0.070	0.077
Labor situation in your main work	−0.045	−0.046	−0.013	−0.010	0.006
Do you have family responsibilities?	0.048	0.097 **	0.040	0.067	0.026
How long have you been working (years) in your present profession?	−0.043	−0.105 **	−0.075 *	−0.115 **	−0.037
Is shift work part of your job?	−0.043	0.030	0.045	0.089 *	0.029
Have you followed specific training in research over the past 5 years?	−0.169 **	−0.274 **	−0.132 **	−0.113 **	−0.139 **
How many courses have you completed?	−0.935 **	−0.946 **	−0.756 *	−0.948 **	−0.736

* Correlation with a significance level of 0.05 (two-tailed); ** correlation with a significance level of 0.01 (two-tailed).

**Table 3 ijerph-19-03821-t003:** Descriptive analysis by professional category on the subscales of the HS-EBP 30 questionnaire (mean and standard deviation).

Professionals	Beliefs		Results		Development		Evaluation		Barriers/Facilit.	
	M	SD	M	SD	M	SD	M	SD	M	SD
Nurse	103.6	13.4	93.6	26.1	77.5	12.7	83.4	22.1	64.4	24.4
Physiotherapist	99.6	16.1	96.6	24.8	79.9	10.2	84.6	21.7	62.5	22.1
Speech therapist	104.9	12.4	106.1	18.9	83.6	9.9	96.5	15.8	72.7	24.5
Others	97.7	23.9	102.6	26.6	81.0	10.6	92.2	15.4	70.4	24.2
Podologist	98.2	13.7	102.7	23.8	83.1	9.9	84.7	22.5	74.9	17.4
Occupational Therapist	101.6	14.9	97.3	21.2	81.3	12.8	94.1	19.4	54.9	25.0

Barriers/facilit. = Barriers and facilitators.

**Table 4 ijerph-19-03821-t004:** Single-factor ANOVA and posthoc tests for the HS-EBP 30 questionnaire subscales by professional category.

Factors	F	*p* *	Partial Eta	Observed Potential
Beliefs and Attitudes	2.95	0.01	0.02	0.86
Results	4.60	0.00	0.03	0.97
Development	5.35	0.00	0.04	0.99
Evaluation	9.33	0.00	0.06	1.00
Barriers/Facilitators	6.78	0.00	0.04	1.00

* significance level *p* < 0.05.

**Table 5 ijerph-19-03821-t005:** Descriptive and inferential analysis of the subscales (HS-EBP 30) between the pre- and post-lockdown groups due to COVID-19.

COVID-19									
Factors	Total		Pre-Lockdown			Post-Lockdown			*p **
	*n*	M (Std. Dev.)	*n*	M	DE	*n*	M	DE	
Beliefs and Attitudes	730	102.38 (14.6)	387	101.1	16.07	343	103.8	12.53	0.011 *
Results	730	96.93 (24.6)	387	95.9	25.26	343	98.1	23.75	0.222
Development	730	79.60 (11.9)	387	79.0	12.54	343	80.3	11.29	0.151
Evaluation	730	87.27 (21.3)	387	85.2	21.66	343	89.6	20.68	0.005 *
Barriers/Facilitators	730	64.27 (24.4)	387	63.4	24.70	343	65.3	24.03	0.299

* Single-factor ANOVA test, at a significance level of *p* < 0.05.

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
