# Peer review of "The Influence of the COVID-19 Pandemic on the Clinical Application of Evidence-Based Practice in Health Science Professionals"

_ijerph, 2022, doi:10.3390/ijerph19073821_

Round 1
Reviewer 1 Report
The authors of presented an important topic about health professional beliefs and attitudes regarding evidence-based practice. There are a few areas that could be strengthened.
- The introduction could include more description of an actual problem - has there been a lack of EBP resulting in poorer patient outcomes for example? There are nursing care guidelines that should be followed per hospital policy - understanding one's beliefs and attitudes would be prompted by identifying increased errors/poor outcomes in a particular area. Also, what is the theoretical foundation for the proposed relationships and analysis?
- Methods: There are no psychometrics provided for the measures, it is unclear why physicians were excluded and other health care workers were not - what was the rationale for the selection. Were participants compensated for their time?
- What are "results" in Tables 2 and 3? Without clear results it is challenging to follow the discussion.
Thank you for the opportunity to review this work.
Author Response
Thank you for taking the time to review our manuscript. We are very grateful for the reviewers’ comments on our paper. We have considered them with care, and the comments have been valuable for us when improving the manuscript. Please find below the comments, our responses and changes made.
The authors of presented an important topic about health professional beliefs and attitudes regarding evidence-based practice. There are a few areas that could be strengthened.
- The introduction could include more description of an actual problem - has there been a lack of EBP resulting in poorer patient outcomes for example? There are nursing care guidelines that should be followed per hospital policy - understanding one's beliefs and attitudes would be prompted by identifying increased errors/poor outcomes in a particular area. Also, what is the theoretical foundation for the proposed relationships and analysis?
Authors: Indeed, there is currently a lot of literature on the subject, we have added some sections expanding on this point, thank you for the suggestion.
- Methods: There are no psychometrics provided for the measures, it is unclear why physicians were excluded and other health care workers were not - what was the rationale for the selection. Were participants compensated for their time?
The theoretical framework underpinning the study, as explained above, is the fact that in Spain there are no specific courses on EBP or formal training. We believe that this is a barrier to professionals researching or consuming scientific literature. This is not the case with physicians, whose curricula include research. Let us bear in mind that in Spain, until a few years ago, the studies of nurses, physiotherapists, speech therapists, etc., lasted 3 years, while the training of physicians lasted 6 years. In addition, doctors could access doctoral studies while other professionals could not, at least not directly from the career. This is the justification for the criteria of choice.
Authors: Participation was voluntary and they were not remunerated in any way.
- What are "results" in Tables 2 and 3? Without clear results it is challenging to follow the discussion.
Authors: Thanks for the suggestion, we have expanded the highlights of each table to improve the reader's understanding.
Reviewer 2 Report
Article review: ijerph-1579869
Thank you for inviting me to review this manuscript. Overall, it is well written and the purpose of the article was appropriate. This is worthy of publishing minus these issues highlighted below.
Issues identified.
Page 2 line 45: delete The in front of Evidence…..
Page 2 lines 58-62. This whole paragraph is written in a way that few of the intended concepts are understood. Rewrite for clarity.
Page 2, line 79. Professionals …. Should start a new paragraph because prior to this sentence you discuss the benefits of EBP. Therefore the barriers should start a new paragraph.
Page 2, line 82. Please explain what investigation was undertaken. Your investigation?
Some references I could not validate as I do not read Languages other than English.
Reference number 21, page 11. Practice is spelled wrong.
Author Response
Thank you for taking the time to review our manuscript. We are very grateful for the reviewers’ comments on our paper. We have considered them with care, and the comments have been valuable for us when improving the manuscript. Please find below the comments, our responses and changes made.
Page 2 line 45: delete The in front of Evidence…..
Page 2 lines 58-62. This whole paragraph is written in a way that few of the intended concepts are understood. Rewrite for clarity.
Page 2, line 79. Professionals …. Should start a new paragraph because prior to this sentence you discuss the benefits of EBP. Therefore the barriers should start a new paragraph.
Page 2, line 82. Please explain what investigation was undertaken. Your investigation?
Some references I could not validate as I do not read Languages other than English.
Reference number 21, page 11. Practice is spelled wrong.
Authors: Thank you very much, we have corrected everything indicated by the reviewer.
Reviewer 3 Report
the title “Barriers, Beliefs and Attitudes of Health Science Professionals in the Clinical Application of Evidence-Based Practice: a Multi-Centric Study (Evidencare Project) is incongruent with the objectives and the type of study.
The introduction only describes Evidence-Based Practice. It doesn't present any evidence of the relationship with COvid 19.
The objective, evaluating the influence of the Covid-19 pandemic on the barriers to, beliefs and attitudes towards EBP is not congruence with the Introduction.
The type of study is not adequate. Consider review.
Table 1 is not correct in different dimensions.
Discussion is poor and centres on Evidence-Based Practice
The Limitation “Initially, Latin-American countries were included in the project, nevertheless, they were finally omitted.” is not compressive.
Conclusions are poor consider review.
Author Response
Thank you for taking the time to review our manuscript. We are very grateful for the reviewers’ comments on our paper. We have considered them with care, and the comments have been valuable for us when improving the manuscript. Please find below the comments, our responses and changes made.
the title “Barriers, Beliefs and Attitudes of Health Science Professionals in the Clinical Application of Evidence-Based Practice: a Multi-Centric Study (Evidencare Project) is incongruent with the objectives and the type of study.
Authors: We have adjusted the title to the objectives, thank you
The introduction only describes Evidence-Based Practice. It doesn't present any evidence of the relationship with COvid 19.
Authors: This is explained at the end of the introduction.
In the course of data collection for this work, a state of emergency was declared and a three-month domestic national lockdown was imposed in Spain, between March and June 2020, due to the Covid-19 pandemic. These events even prompted the enlargement of the study, incorporating a second objective: to evaluate the influence of the Covid-19 pandemic on the barriers to, beliefs in and attitudes towards EBP among university professionals of health sciences, excluding doctors.
The objective, evaluating the influence of the Covid-19 pandemic on the barriers to, beliefs and attitudes towards EBP is not congruence with the Introduction.
The type of study is not adequate. Consider review.
Authors: We do not understand, what do you mean? ? If you could be so kind as to be more specific, we could correct it.
Table 1 is not correct in different dimensions.
Authors: We do not understand, what dimensions are you referring to? If you could be so kind as to be more specific, we could correct it.
Discussion is poor and centres on Evidence-Based Practice
Authors: We briefly discussed the impact of COVID19 in part, but it is difficult to do so at greater length because there is little comparative literature.
The global pandemic due to Covid-19 has become a source of great stress among health professionals. It motivated us to perform a comparative pre- and post-lockdown at home with the aim of showing possible differences with regard to interest in EBP. In our study, the post-lockdown group showed a significative change in the variables related with the two blocks “beliefs and attitudes” and “evaluation”. A significative increase in the interest of interviewees towards these areas of EBP, possibly justified by the need for constant knowledge updates and for applying evidence, given the high uncertainty generated by the virus.
The Limitation “Initially, Latin-American countries were included in the project, nevertheless, they were finally omitted.” is not compressive.
Authors: We have indeed corrected it, thank you.
Conclusions are poor consider review.
Authors: We have rewritten this section
Reviewer 4 Report
Thanks to the authors for letting me review this study. Congratulations for the effort made. First of all, I wanted to add that the title is too long and as advice I would give it a spin. The authors should review the methodology, the introduction and the results... Also the English.
Author Response
Thank you for taking the time to review our manuscript. We are very grateful for the reviewers’ comments on our paper. We have considered them with care, and the comments have been valuable for us when improving the manuscript. Please find below the comments, our responses and changes made.
Thanks to the authors for letting me review this study. Congratulations for the effort made. First of all, I wanted to add that the title is too long and as advice I would give it a spin. The authors should review the methodology, the introduction and the results... Also the English.
Authors: Thank you very much, we have corrected everything indicated by the reviewer. We have adjusted the title to the objectives.
Reviewer 5 Report
Thank you for allowing me to review this work. It is very interesting, although it needs some minor clarifications or adjustments. These are detailed below.
Methods: There is a lack of data on how the questionnaire dissemination and sample selection was done. Were any completed questionnaires deleted, and why?
The Data Protection Act cited 1999 is not in force, it was replaced by a later law of 2018. Check
Results: There is a normal distribution of variables? from the analyses presented it seems so, but this should be clarified.
Why have they included variables such as gender (qualitative) in a correlation? This should be clarified.
Author Response
Thank you for taking the time to review our manuscript. We are very grateful for the reviewers’ comments on our paper. We have considered them with care, and the comments have been valuable for us when improving the manuscript. Please find below the comments, our responses and changes made.
Methods: There is a lack of data on how the questionnaire dissemination and sample selection was done. Were any completed questionnaires deleted, and why?
Authors: This is a good point. All questionnaires were 100% completed. A purposive sample selection was made, accepting all the questionnaires that were received. No questionnaires were eliminated since they were all complete.
The Data Protection Act cited 1999 is not in force, it was replaced by a later law of 2018. Check
Authors: Indeed, we have corrected it, thank you very much.
Results: There is a normal distribution of variables? from the analyses presented it seems so, but this should be clarified.
Authors: In the Statistical Analysis section we indicate that the following was performed: The Shapiro-Wilk normality test, Levene’s Homogeneity of Variances
Why have they included variables such as gender (qualitative) in a correlation? This should be clarified.
Authors: The inclusion of the gender variable (even though it is qualitative, Pearson's r can be used, since most of the variables are quantitative) is common practice, although not relevant, in this type of study.